# A Recommender System for Increasing Energy Efficiency of Solar-Powered Smart Homes

**DOI:** 10.3390/s23187974

**Published:** 2023-09-19

**Authors:** Quentin Meteier, Mira El Kamali, Leonardo Angelini, Omar Abou Khaled

**Affiliations:** 1HumanTech Institute, University of Applied Sciences and Arts Western Switzerland (HES-SO), 1700 Fribourg, Switzerland; quentin.meteier@hes-so.ch (Q.M.); mira.elkamali@hes-so.ch (M.E.K.); omar.aboukhaled@hes-so.ch (O.A.K.); 2School of Management Fribourg, University of Applied Sciences and Arts Western Switzerland (HES-SO), 1700 Fribourg, Switzerland

**Keywords:** energy consumption, home automation, machine learning, recommender system, smart home, solar production, user habits

## Abstract

Photovoltaic installations can be environmentally beneficial to a greater or lesser extent, depending on the conditions. If the energy produced is not used, it is redirected to the grid, otherwise a battery with a high ecological footprint is needed to store it. To alleviate this problem, an innovative recommender system is proposed for residents of smart homes equipped with battery-free solar panels to optimise the energy produced. Using artificial intelligence, the system is designed to predict the energy produced and consumed for the day ahead using three data sources: sensor logs from the home automation solution, data collected by the solar inverter, and weather data. Based on these predictions, recommendations are then generated and ranked by relevance. Data collected over 76 days were used to train two variants of the system, considering or without considering energy consumption. Recommendations selected by the system over 14 days were randomly picked to be evaluated for relevance, ranking, and diversity by 11 people. The results show that it is difficult to predict residents’ consumption based solely on sensor logs. On average, respondents reported that 74% of the recommendations were relevant, while the values contained in them (i.e., accuracy of times of day and kW energy) were accurate in 66% (variant 1) and 77% of cases (variant 2). Also, the ranking of the recommendations was considered logical in 91% and 88% of cases. Overall, residents of such solar-powered smart homes might be willing to use such a system to optimise the energy produced. However, further research should be conducted to improve the accuracy of the values contained in the recommendations.

## 1. Introduction

Since 2010, photovoltaics have been steadily increasing their share of cumulative electricity capacity worldwide. In 2023, they occupy third place, with a share of 14.7%, behind coal (24.7%) and hydroelectricity (15.7%) [1]. In recent years, both companies and private households have installed photovoltaic systems for various reasons. This is firstly for financial reasons, as it reduces the cost of energy consumed once the cost of installation has been absorbed [2]. But it is also attractive in the short term, given the tax breaks introduced by governments to encourage people to take the plunge. Secondly, people are also doing it for ecological reasons [2]. In the long term, the use of solar energy has a much smaller carbon footprint than electricity produced with fossil fuels. Given the ecological emergency faced by our planet but also in order to diversify energy sources in the face of the current energy crisis, the number of photovoltaic installations should continue to rise.

There are two types of installation: those with a battery and those without one. Having a battery increases the home’s autonomy, as it enables the energy produced to be stored and used at any time of day. However, the production of a battery has a high carbon footprint, as it uses rare materials that will soon be exhausted, and is often quite expensive. It would, therefore, be preferable to opt for battery-free systems while optimising the instant use of the solar energy produced.

However, optimising the use of the solar energy produced can be complicated for the homeowner. They should schedule tasks according to personal and private schedules, without having access to consumption and production data. Such a task can be handled by an artificial intelligence system, which can plan the execution of energy-intensive household tasks according to a number of parameters: weather and temperature, energy consumed and produced, occupant habits, etc. A recommender system could be appropriate in this context: it would help the end-user to make decisions and optimise the energy produced while leaving him/her free to choose whether or not to apply the recommendation. Indeed, a user would prefer an automated system that gives them recommendations and over which they have control, rather than a totally autonomous system that performs tasks for them [3]. To maximise the chances of users following recommendations, the latter would need to be adapted to their habits. In fact, a system that takes into account the context (i.e., the user’s habits) should suggest recommendations that are more attractive and better adapted to the user’s mood and tastes. This should have a positive impact on the acceptance and thus the use of such a system [4]. That is the reason why including a home automation solution in the system architecture would help to better understand the user’s habits and provide recommendations tailored to their needs. If so, this could have a beneficial economic and ecological impact in the long term for many inhabitants equipped with solar panels. Additionally, proposing data visualisation for the end-user might increase their awareness of energy consumption and production [5].

The main innovation of the proposed system is to propose recommendations to the resident using information provided by non-intrusive sensors in a home automation solution, in addition to weather and solar inverter data. This would give us a better understanding of the user’s consumption habits and activities, with the aim of making increasingly relevant recommendations. The other difference is that the system is trained on data from an individual home equipped with solar panels but without a battery to store energy, to further reduce the carbon footprint of the inhabitant.

This manuscript details the design, implementation, and evaluation of such a recommender system. The data used to train and evaluate the system are those of a Swiss resident equipped with solar panels and a home automation solution.

## 2. Related Work

### 2.1. Prediction of Energy Consumption and Production Using Machine Learning

Most previous work to predict energy consumption and production with data-driven approaches was performed for buildings comprising several apartments/households [6,7,8,9,10]. White-box and grey-box approaches were originally used, but the emergence of big data had research move towards large-scale black-box approaches [7]. Previous studies that attempted to predict energy consumption in buildings varied a lot in terms of number of buildings considered, data sources/features, sampling frequency, duration of data collection, and algorithms [6]. No meta-analysis was conducted, which does not help to choose the right parameters and features to accurately predict energy consumption. With machine learning (ML) algorithms, K-nearest neighbours and support vector machines were shown to achieve high accuracy [9]. Yet, the pattern to be predicted in that study seemed very similar from one day to the next, which must have made the prediction easier. Now, deep learning (DL) algorithms seem to outperform ML algorithms in predicting energy consumption [10], with features describing rooms being among the most predictive ones. Additionally, transformers combined with stationary wavelet transforms might also be a reliable alternative for predicting energy consumption in households [8].

Regarding solar energy production, it can be accurately predicted from solar irradiance from past days using neural networks [11]. Additionally, it can also be accurately predicted from solar radiation combined with other weather data (temperature, wind direction and speed, humidity, pressure, etc.), with both ML and DL [12,13]. Hybrid forecasting (i.e., combining different algorithms for different steps of the prediction pipeline) and recurrent neural networks (RNNs) are now mostly employed [14].

### 2.2. Recommender Systems for Energy Saving in Smart Homes

Himeur et al. [5] proposed a comprehensive review of recommender systems for energy efficiency in buildings. They proposed a taxonomy with categories such as objectives, computing platforms, recommender models, main stages’ incentive measures, and evaluation metrics.

Some propositions have been made in the previous literature for systems related to energy saving specifically in smart homes [15,16,17]. To provide accurate recommendations to save energy in a smart home, activity detection coupled with a rule-based energy-saving framework was implemented by Alhamoud et al. [15]. Users’ activities were detected accurately, but the method includes the use of smart meters and motion sensors, which implies a high level of intrusiveness into the private lives of residents. Also, the logic for implementing the suggestions (rules implemented, frequency of sending) is not clearly reported in the manuscript. Dhage et al. [16] could predict the hourly produced solar energy with ML. Based on that, they provided a first idea to plan recommendations for the user to use it efficiently. However, they are not end-to-end recommendations involving smart planning.

Thus, the system proposed in this manuscript aims to propose a new approach that could solve these problems. The aim is to generate recommendations that plan activities to efficiently use the solar energy produced according to the user’s habits while respecting their privacy, without installing smart meters and motion sensors.

## 3. Design and Implementation

First, the infrastructure and method for data collection are detailed. It was chosen to collect our own data due to a lack of benchmark datasets [5]. Before getting into the description of the design and implementation of the recommendation pipeline, data exploration and ML experiments carried out to predict energy production and consumption are described. This is because the best ML models are then used for inference in the recommendation pipeline.

### 3.1. Data Collection and Exploration

#### 3.1.1. Hardware Infrastructure

Figure 1 shows the overall infrastructure that was set up at the customer’s smart home to collect the data needed to train the recommender system. It consists of collecting three data sources: (1) the logs of the home automation solution, (2) the energy consumed and produced collected with a solar inverter, and (3) the weather data collected from an application programming interface (API). The smart home solution, the solar inverter, and the server collecting and processing the data (LIVA Z3 PLUS, Intel Core i7-10510U CPU @ 1.8 GHz, 16 GB RAM, Ubuntu 22.04 LTS 64-bit) were all connected to the customer’s local network. This infrastructure is meant to be used in edge computing, without data being transmitted to cloud AI for privacy reasons.

The home automation solution comprised devices connected to a smart home module (Control 4 brand). The full list of connected devices, which were mostly lights, can be found in Table A1. All the logs were saved in a text file on the smart home module and accessed by the server through the network.

The solar panels (S79Sol, Aleo Solar GmbH, Prenzlau, Germany) nominal power of 310 Wp) installed consisted of 12 monocrystalline photovoltaic modules covering a surface of 20.77 m${}^{2}$, for the total power of 3.72 kWp and the estimated energy production of 3557 kWh/year on the day of installation. It was orientated towards south-east (−45°) with a pose angle of 37°. The solar inverter (Plenticore Plus 4.2, KOSTAL Solar Electric GmbH, Freiburg im Breisgau, Germany) was connected to the solar panels and to the local server.

#### 3.1.2. Method for Data Collection

A routine was implemented to automate data collection using the *crontab* Linux command on the server. Data were collected from 27 July to 11 October, i.e., 76 days of data (1 day missing). The sampling data frequency was set to 5 min to obtain fine-grained information to perform the ML tasks while maintaining a reasonable number of data [6]. Regarding the device logs, each row followed the following format:

DD/MM/YYYY HH:MM:SS - *<DeviceType><Room><DeviceName>* is <ON/OFF>

Every day, a routine copied the content of the file of the last day on the server and processed the data to fit the sampling frequency. For each device and each 5 min slot, the number of events (ON/OFF) and the ignition time were calculated. A Python script was executed every 5 min for energy consumption and production to trigger a read-only query on the solar inverter through the TCP/IP Modbus protocol [18]. A set of 117 features were collected, including the total home consumption from the solar panels and from the grid, and the solar energy produced. Additionally, weather data were collected from the Visual Crossing API [19] for the current day. The list of features can be found in Table A2. Data were obtained hour per hour and were also converted to fit the sampling data.

#### 3.1.3. Data Exploration

##### Logs from the Home Automation Solution

Given that solar panels produce energy during the day, it was decided to analyse the proportion of time during which lights were switched on between day and night (i.e., before sunrise and after sunset). The light logs of certain rooms were grouped together, and those that were turned on for less than 0.5% of the day were removed from the analysis. The latter revealed that all the lights in the house were turned on for less than 8% of the day. It suggests that it might be difficult to use this source of data to predict the energy consumed, which was confirmed through ML experiments (see Section 3.2.2 below). Logically, it was also found that the lights were turned on more often in the morning and evening.

##### Energy Consumption and Production

The exploratory analysis revealed that most of the energy consumed was consumed by the boiler with an integrated heat pump, which consumed around 3 kW in 1 h every 3 h. As a boiler’s consumption is quite high compared with other household appliances, it is possible that an ML model could base its predictions on the boiler too much and thus be biased. Figure 2 shows the energy consumed and produced in the house during the considered period. The regression lines in the figure show that the energy produced decreased with time, while the energy consumed increased. This pattern makes sense, as we moved from summer to autumn, with decreasing daytime, less sun, and decreasing temperature. We can also observe that the energy consumed in the house could not have been covered by the energy produced by the panels, even if the solar installation had included a battery. This reinforces the relevance of the proposed recommender system to cover as much consumed energy as possible with the produced energy.

### 3.2. Prediction of Energy Consumption and Production Using Machine Learning

ML techniques were used to perform regression tasks on the collected data to predict energy consumption and production. Several experiments were performed to avoid biases in the predictions and achieve the lowest error possible. The preprocessing and training steps common to all experiments are detailed in Section 3.2.1, while the aim, specificities, and results obtained for each experiment are described in Section 3.2.2 and Section 3.2.3. The scikit-learn and Keras frameworks in Python were used for implementation.

#### 3.2.1. Preprocessing and Training Models

Null values were dropped, and features were separated from the ground truth. Then, categorical features were encoded, and two features were extracted from the raw data: the time and the month. All features were normalised between the minimum and maximum values of data distribution (MinMaxScaler). Three different techniques were tested for selecting features: SelectKBest, RFECV, and ExtraTreesRegressor. Different regression algorithms were trained in the experiments: a passive aggressive regressor (PA), a stochastic gradient descent regressor (SGD), a multilayer perceptron (MLP), a single decision tree (DT), an ensemble of decision trees (“Extra-Trees”, ETs), and linear regression (LR). The dataset was split in a training set (60%), a validation set (20%), and a test set (20%). A grid search approach was employed to tune the hyperparameters of the algorithms on the training and validation sets, before being evaluated on the test set with the R${}^{2}$ score and the Root Mean Squared Error (RMSE) as metrics. All the values tested for each hyperparameter of each algorithm in the grid search process can be found in Table A3.

#### 3.2.2. Prediction of Energy Consumption

The experiments conducted are summarized in Table 1, while Table 2 presents the results obtained. The initial idea was to predict the energy consumed based on the sensor logs in the house, as it would have given information on the user’s habits (EC1). Yet, as suggested by the data exploration, the predictions were not accurate. The same task was thus tried with weather data as inputs (EC2). High accuracy was achieved, but the model evaluation was biased because of the data leakage phenomenon [20]. Indeed, the whole dataset was shuffled before splitting it, so data from the same day were both in the training/validation and test sets. This problem was addressed in further experiments. The EC3 experiment was similar to EC2 but without shuffling the dataset. It was similar to the task that would be performed in real situations, i.e., predicting the energy consumed for the coming day for which data are unknown to the model. Because the results were not conclusive, we tried to predict the cumulated energy consumed instead of predicting the instant energy consumed (EC4). The model accuracy was higher, but when transforming the predictions into instant energy consumed, we figured out that predictions were far from the ground truth. Then, we tried to predict the energy consumed every hour instead of every 5 min (EC5), which gave the most reliable and accurate predictions (R${}^{2}$ score of 13.56%), although this is far from a perfect prediction. Results are further discussed in Section 5.2.

#### 3.2.3. Prediction of Energy Production

Similarly to energy consumption, three experiments were conducted for predicting the energy produced with solar panels. They are summarized in Table 3, while the results are presented in Table 4. The initial idea (EP1) was to predict the energy produced using weather data, the time of the day, and the month. A very low error was achieved, but a similar bias for the prediction of the energy consumed was observed. Thus, a second experiment (EP2) was conducted on the same input features without shuffling the data and keeping the dataset ordered by date. It gave the most accurate prediction of the energy produced, with an R${}^{2}$ score of 84.81%. A third experiment was conducted (EP3) to investigate if the model would have benefitted from resampling the data and predicting the solar energy produced hourly rather than every 5 min, but it was not the case (R${}^{2}$ score of 82.27%).

### 3.3. The Recommender System

#### 3.3.1. Recommendation Pipeline

The goal of the recommender system is to suggest that the resident perform energy-intensive household tasks (called *activities* in this manuscript) at specific times of the day, depending on the solar energy being produced and the user’s energy consumption habits. The pipeline executed to generate the recommendations is shown in Figure 3. It was run early in the morning each day (e.g., at 2 a.m.). The modules/functions implemented and run to generate the recommendations are explained below. In the final product, the user would receive recommendations on his/her smartphone every morning (e.g., at 8 a.m.).

The *Consumption habits* module represents the user’s habits in terms of household activities. The *Consumption and Production prediction* modules generate the predictions of energy consumption and production for the day ahead. To do that, ML models were trained on retrieved data (see Section 3.2), and the best ones were used for inference. The predictions of consumed and produced energy are compared to find out the time(s) when there is unused solar energy. The system tries to place the activities to be carried out at these time(s), depending on the user’s habits. Based on that, textual recommendations that associate activities to perform and a time of the day (e.g., “Do your laundry at 1:30 p.m. today because...”) are generated.

Additionally, the *Recommendations unrelated to user habits* module gathers recommendations that are not based on the predictions of consumed and produced energy. They are either facts about appliance consumption (showers, hobs, etc.) to increase the user’s ecological awareness, advice on heating according to the temperature forecast, or positive/negative comments about the use of the solar energy produced on the previous day. These more general recommendations were implemented in the system to offer diversified recommendations, which is a key factor for user acceptance [21]. The recommendations are then ranked using a multi-criteria decision-making process such as Technique for Order of Preference by Similarity to Ideal Solution (TOPSIS) [22].

#### 3.3.2. Consumption Habits

The sensor logs collected in the house did not allow us to link the use of household appliances to the energy consumed (see experiment EC1 in Section 3.2.2). Thus, it was chosen to simulate the user’s habits using a JSON file as shown in Listing 1. Each activity is defined by *appliance*, *frequency* at which it is performed (in days), its *duration* (in hours), the time *range* during which it is carried out, the electrical *power* theoretically consumed (in kW), and the number of days it has not been included in a recommendation (*last_recommended_in_days*). Every day in which the activity is not included in a recommendation, this last variable is incremented. If an activity is selected by the system to be recommended, *last_recommended_in_days* must be equal to or greater than the specified *frequency_in_days*.

**Listing 1.** JSON file simulating activities carried out by the user.

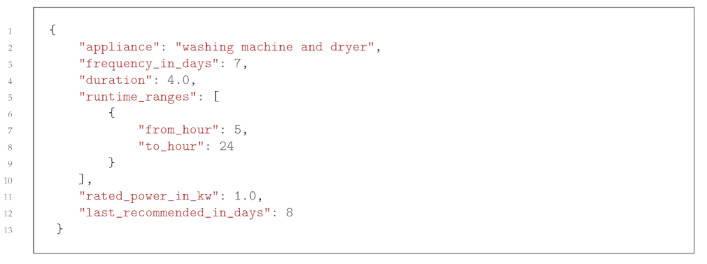



#### 3.3.3. Generating Recommendations

##### Recommendations Related to the User’s Habits

The consumption and production predictions are first compared to create two signals: one representing the estimated overconsumption (in blue in Figure 4) and one representing the estimated solar energy available (i.e., without the predicted consumed energy; in pink in Figure 4). Both are smoothed using linear convolution to simplify the search for peaks.

Another module tries to place the activities to be suggested by the recommender system under the curve where production is high enough. More precisely, the system analyses the width (i.e., duration of the activity) and the height (i.e., the energy needed) for each production peak contained in Figure 4. If an activity fits a peak, the system proposes a recommendation to perform such activity at the peak time. Otherwise, it simply creates a recommendation explaining the amount of solar energy available.

For a better understanding of how this works, here is a concrete example for the day of 25 August 2022: Figure 5 shows energy consumption and production on that day. In particular, the orange dashed line shows the produced energy available throughout the day. It is at times when this curve is high that the system attempts to fictitiously place activities, as shown in Figure 6. We can see 3 activities that could be placed by the system, each of which would then correspond to a recommendation: hobs in yellow, the dishwasher in green, and available energy (not associated with any appliance) in magenta. The brown signal is the actual consumption to which we have added the recommended activities. This signal shows that global consumption would not increase much while using the energy produced and previously unused.

##### Recommendations Unrelated to the User’s Habits

This module generates recommendations unrelated to the user’s activities, which are also structured in a JSON file. It contains all the recommendations that can be selected by the system, and an example of one recommendation is shown in Listing 2. Each is defined by its *type*, its plain *text*, the *condition* under which it is selected, the energy *saving* generated if it is applied (in kW), the *likelihood* that the user will apply it (in %), the *frequency* at which it can be selected (in days), and the number of days in which it has not been selected (*last_recommended_in_days*).

**Listing 2.** JSON file formatting recommendations not related to energy consumption and production.

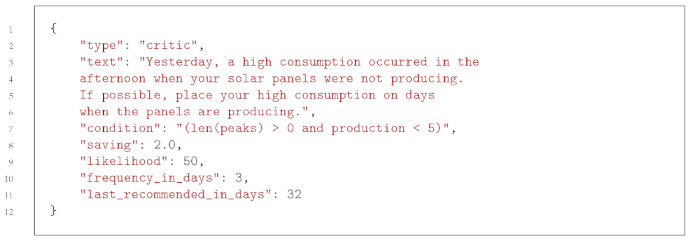



#### 3.3.4. Recommendation Ranking

Usual recommender systems only take into account a user’s preferences for an item to satisfy him or her. We write this as an *f* function, with *Rating* being the output value we want to maximize to generate the best recommendation possible:(1)f:User×Item⇒Rating

In our case, the ranking assigned to a recommendation depends on more parameters than simply the match between the user and the best item. In this case, we want to suggest that the user perform certain energy-intensive *activities* at the best *time* of the day (i.e., when solar production is high). However, some activities are difficult to carry out at any time of day, particularly depending on weather conditions, which makes the *context* another important factor to take into account. In addition, a good recommendation should *save* the user energy (and money). For this to happen, the *likelihood* of the user applying the recommendation must be high, especially in relation to their routine and consumption habits. Thus, the evaluation function becomes the following:(2)f:Activity×ContextTime×Saving×Likelihood⇒Ranking

Table 5 details the meaning of each variable, with its type and potential range/unit. The above function was implemented and used in the TOPSIS ranking technique to assign a rank to each recommendation.

## 4. Evaluation

### 4.1. Method

Two variants of the recommender system were implemented and evaluated:Variant 1, taking into account the predicted energy to be consumed.Variant 2, without taking the predicted energy to be consumed into account (based solely on energy production predictions).
The second variant was created to investigate whether recommendations based solely on energy production would satisfy the user equally well. For each day of the data collection periods, recommendations were generated, and those selected from 15 to 28 August 2022 (14 days) were used for system evaluation. All the recommendations that could potentially be selected by the system each day appear in Table 6 (*Textual Recommendation* column).

Eleven people who were not residents of the household where data were collected participated in the evaluation process. The results are presented in Table 6. They had to fill in a template document to evaluate the relevance of the recommendations (i.e., textual content), the predicted values they contained (time of day and energy in kW), and the ranking. Firstly, the document presented all the recommendations, and respondents were asked to tick the “Relevant” column if they would have liked to have received such a recommendation (based on the raw text). Then, for each day and for both variants, the document presented the recommendations selected by the system, arranged in ascending order based on TOPSIS ranking, along with a graph like the one in Figure 7. Respondents had to answer in a two-choice column (yes vs. no) whether the values contained in the textual content of the recommendations (time of day and energy in kW) made sense regarding the data prediction presented in the graph. They were also asked to tick the “Logical” column if they thought that the ranking of recommendations made sense to them.

Apart from the evaluation process conducted by the respondents, the diversity of the recommendations selected by the system was also investigated. For each recommendation, the number of times it was selected over the 14 days is shown in Table 6 (see *Occurrences* column).

### 4.2. Relevance of Recommendations

The results (*Relevance* column in Table 6) are presented as a percentage, with 100% indicating that all participants would have liked to receive this recommendation. It shows that the recommendations were rated relevant in 74% of cases on average. Seven were appreciated by more than 80% of the respondents. However, three recommendations were considered not relevant by more than half of the respondents (less than 50% of relevance), so they could be adapted or removed in the next version of the system.

### 4.3. Accuracy of Values Contained in Recommendations

Regarding the values contained in the recommendations (*Values* column in Table 6), the respondents reported that they were accurate in 66% and 77% of recommendations selected by variants 1 and 2 of the system, respectively. For variant 1, the time predictions of boiler activation were usually offset, which explains the low accuracy for that recommendation (38%). Respondents surely noticed the discrepancy between the times in the text and those corresponding to consumption peaks in the graph. If we remove this one to calculate the average accuracy of values, the accuracy rises to 73% for variant 1.

### 4.4. Diversity of Recommendations

Variant 1 of the system selected 56 recommendations over the 14 days (4 recommendations per day on average). Three recommendations were selected (almost) every day by the system, with each representing more than 20% of selected recommendations. Five recommendations were selected five times or less by the system over the 14 days, ranging between 2% and 9% of the selected recommendations. Three recommendations were never selected by the system. Two of them are related to bad and cold weather, which is not likely to occur in August. The last one would be selected if the consumption were high and the production were low, which are also not likely to occur in August. Variant 2 selected only 27 recommendations in total (almost 2 per day on average), which is half of those of variant 1. Two recommendations were selected every 2 days, each representing more than 25% of selected recommendations. Three recommendations were selected three times (11% of selected recommendations), and two were selected less than three times. Three recommendations were never selected by the system (the same as variant 1), and one recommendation could not be selected as it was only related to the consumed energy.

### 4.5. Ranking of Recommendations

On average, over the 14 days, the recommendations ranking was rated relevant in 91% and 88% of cases for variants 1 and 2, respectively.

## 5. Discussion

### 5.1. User’s Habits

As suspected according to data exploration and confirmed with the ML tasks, it was not possible to fully use the potential of the logs from the smart home automation solution to understand the user’s habits and in particular to link it with the energy consumed in the house. To better understand the user’s habits regarding energy-intensive household tasks, a solution could be to install smart meters on household electric appliances and connect them to the home automation solution. Not only it would give more relevant information about the habits of the inhabitants, but the consumption for each household appliance might also be easier to predict than the user’s consumption habits with ML techniques [6,7,9]. However, this would make the system more intrusive into the private lives of residents, which could put a brake on the adoption of such a system. Another alternative could be to apply Non-Intrusive Monitoring (NILM) techniques to detect the use of household appliances from the aggregated consumption signal [23]. However, some device activation might be more difficult to detect than others [24].

If the activities related to household appliances can be predicted accurately enough, the user’s habits could be formatted with the same format as simulated in this version of the proposed recommender system (e.g., Listing 1). In this way, the recommender system would be fully automated while fitting the user’s habits.

### 5.2. Prediction of Energy Consumption and Production

For both consumption and production prediction, the models overfitted the collected weather data (EC2 and EP1) when shuffling the dataset (i.e., training and evaluating the models on data collected the same day). This reminds us that researchers should be careful when interpreting results obtained with ML techniques on time-series data.

Additionally, even with an unshuffled dataset, energy consumption could hardly be explained by the models based on sensor logs, as expected after data exploration. It was also hard to predict it based on weather data. The best performance was obtained with the hourly consumed energy as ground truth (R${}^{2}$ score of 13.56%, EC5). Yet, this is not enough for the concrete use of such a system in the wild. Indeed, the RMSE achieved in the different experiments (between 0.57 and 1.75) is higher than in previous work. Saoud et al. [8] obtained RMSEs of 0.004 to 0.009 depending on the time intervals used (5 to 30 min, respectively), while Bashawyah and Qaisar [9] achieved MSEs between 1.08% and 2.4%, which also suggests a lower prediction error. The former used a combination of transformers and stationary wavelet transforms for prediction, while the latter had a much more consequent dataset for training the model (data collected in 5566 households over 3 years). These are two avenues on which we need to focus our research to improve the prediction of consumed energy for our use-case scenario.

Energy production could be accurately predicted by the models. The prediction of hourly produced solar energy was less accurate (R${}^{2}$ score of 82.27% and RMSE of 0.37; EP3) than the 5 min window prediction (R${}^{2}$ score of 84.81% and RMSE of 0.36; EP2). However, the prediction accuracy was lower than in previous works that used similar sources of data (weather data and solar radiation). Jebli et al. [12] achieved R${}^{2}$ scores of 93 to 95% for scenarios that did not overfit, while Al-Jaafreh et al. [13] achieved an RMSE of 0.035 using 16 features to predict the hourly produced energy. Those slightly better results might be explained by the bigger size of datasets and algorithms used (neural network-based architectures). Additionally, Alomari et al. [11] achieved an RMSE of 0.07 using irradiance from the previous five days. The methodologies and algorithms used in these previous works, combined with a longer period of data collection, should be considered to improve our predictions of energy produced.

### 5.3. Relevance and Diversity of the Recommendations

Most of the recommendations were appreciated by the respondents, but three recommendations did not receive great interest. The first one (“We produce enough to power appliances of around 1700 W from 10 a.m. to 11 a.m.”) might have been not specific enough, because no details on which appliance should be used were given. The second one (“We predict that periodic consumption will consume about 3 kW for about an hour at these times: 2 a.m., [..], 0 a.m.”) just gave information about the high periodic consumption (3 kW every three hours) detected by the system. In this case, we suspect that it is the boiler that reactivates every 3 h once the low set temperature (55°) is reached to heat the water for about 1 h and reach the high set temperature (60°). Here, the resident would have to contact a specialist to adjust or resize the water heater. We believe that such a recommendation may be useful to avoid wasting energy and money, but this was not perceived as such by respondents. The third one (“Glass ceramic hobs have a power rating of 2000 W, while induction hobs have a power rating of 2800 W. On the other hand, induction hobs heat up faster, which means they use less energy on average.”) was a generic recommendation comparing ceramic hobs and induction hobs. It was the least preferred by those questioned, probably because it offers no advantage to the end-user in terms of improving their behaviour over the next few days. This type of recommendation will probably have to be removed from the list. The results regarding the relevance of recommendations, whether good or bad, indicate that end-users prefer to receive precise recommendations suggesting which specific device should be used at which time of day. Advice that is too broad and may not have a beneficial impact on their behaviour in the following days should be avoided. This should be taken into account when refining the list of recommendations implemented in the next version of the system while maintaining an appropriate level of diversity.

Speaking of the latter, our system showed that three recommendations were selected more often than the others in each variant of the system. This is not ideal if one of the recommendations is not appreciated by users, as was the case here. However, the system was only evaluated over 14 consecutive days in August, and it seems logical that some recommendations were not selected, as explained above (see Section 4). Diversity should, therefore, be evaluated on recommendations generated over a whole year to limit the effect of seasonality.

### 5.4. Limitations and Further Research

This first version of the system obviously has a few limitations. Unfortunately, the device logs did not help much to link the user’s habits with the energy consumed in the house. Also, one major issue we faced was to understand that the high periodic consumption in the house was due to the boiler, which was oversized in relation to the size of the house. Detecting these recurrent boiler ignitions using time-series analysis techniques and isolating them from the rest of the aggregated consumption signal represent another idea to improve the prediction of the energy consumed.

In this work, the *likelihood* (i.e., the probability that the recommendation will be accepted by the resident) is currently arbitrary. If we manage to detect habits more accurately, we could define and adapt this *likelihood* according to the user’s behaviour over the first few weeks of using the system.

Of course, the results presented are those of the current system, based on data collected from a single household in Switzerland over a limited period. It would be interesting to study whether similar results can be obtained with data collected from other households, of different sizes, for a longer period and in different seasons, in other regions or countries, with a greater variety of inhabitants (a single person vs. a couple vs. a family) who have different habits. Here, the user had no well-defined routines, which certainly made the prediction of energy consumption more complex.

Many adjustments can be made to improve the proposed recommender system, notably by adding other recommendations to increase diversity or by modifying existing recommendations on the basis of the results obtained. In addition, a module for creating the recommendation text using natural language generation (NLG) techniques could be added. Currently, some recommendations already explain how the system arrived at a decision, but users may find that some are not explicit enough. Explainable AI (xAI) is one of the major challenges for recommender systems, particularly in encouraging the adoption of this type of automated system by the general population [5]. It would, therefore, be interesting to investigate the effect of the xAI modality (no xAI vs. textual explanation vs. textual explanation + image) or the level of information (little vs. very detailed recommendation) on users’ intention to use, appreciation, and trust.

## 6. Conclusions

Overall, this proposal of a recommender system for battery-free solar-powered smart homes was appreciated and could enable their residents to optimise the energy produced while leaving the final choice to the user when it comes to carrying out energy-intensive activities in the house. Still, further work should be performed to increase the accuracy of the values contained in the recommendations by collecting more data in other contexts, removing non-appreciated recommendations, and analysing the diversity of recommendations over a longer period.

## Figures and Tables

**Figure 1 sensors-23-07974-f001:**
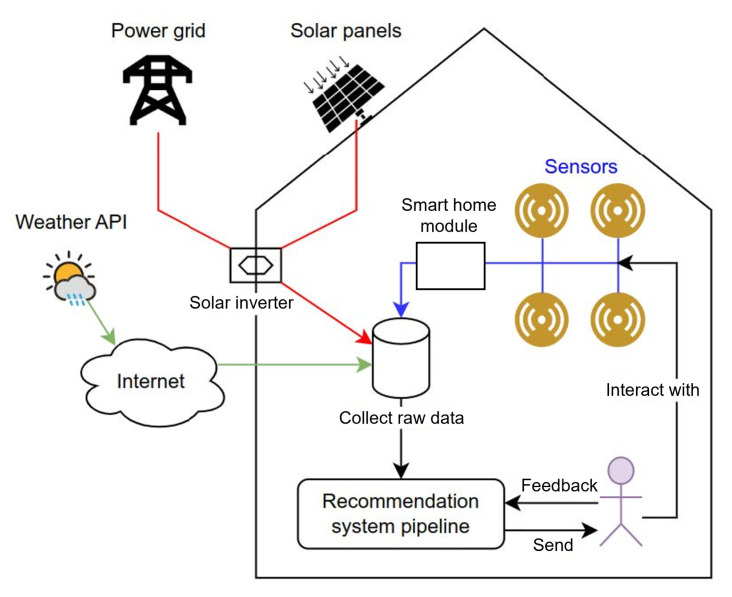
Infrastructure for collecting data and sending recommendations to the user.

**Figure 2 sensors-23-07974-f002:**
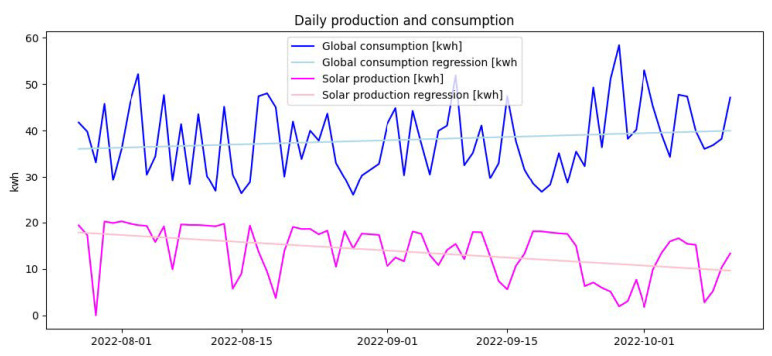
Daily produced and consumed energy over the data collection period.

**Figure 3 sensors-23-07974-f003:**
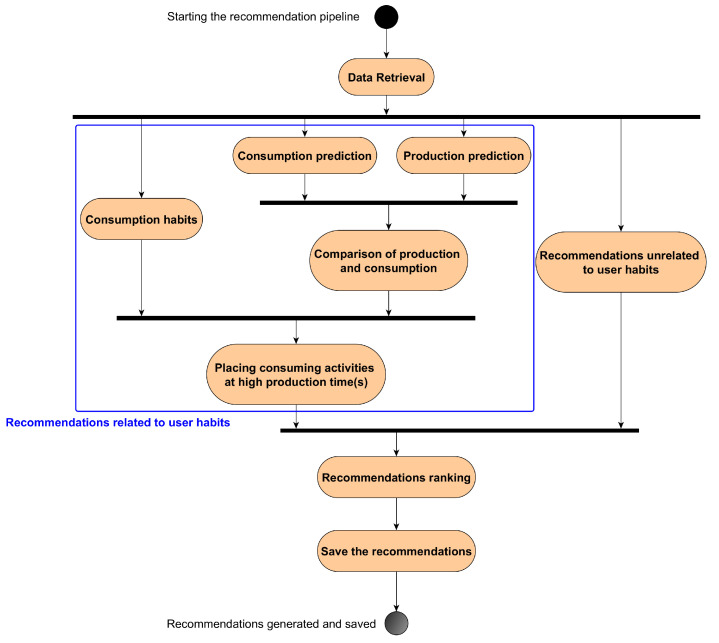
Flow of the recommendation pipeline. The blue square encompasses the modules participating in the generation of recommendations related to user habits.

**Figure 4 sensors-23-07974-f004:**
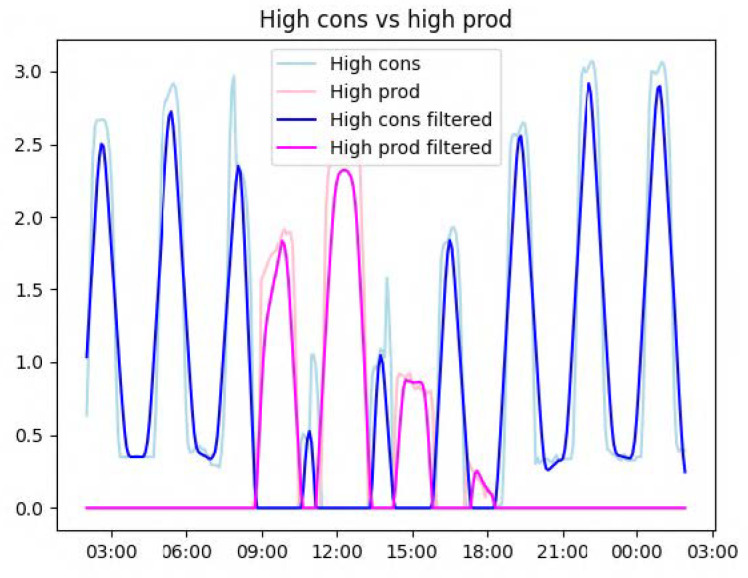
Graph illustrating the comparison process between the energy consumed and produced.

**Figure 5 sensors-23-07974-f005:**
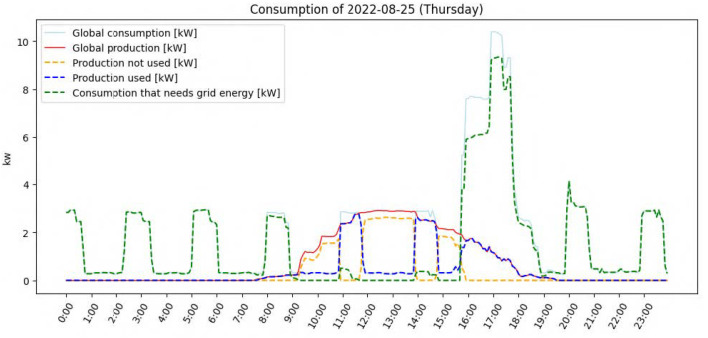
Energy consumption and production in the house on 25 August 2022.

**Figure 6 sensors-23-07974-f006:**
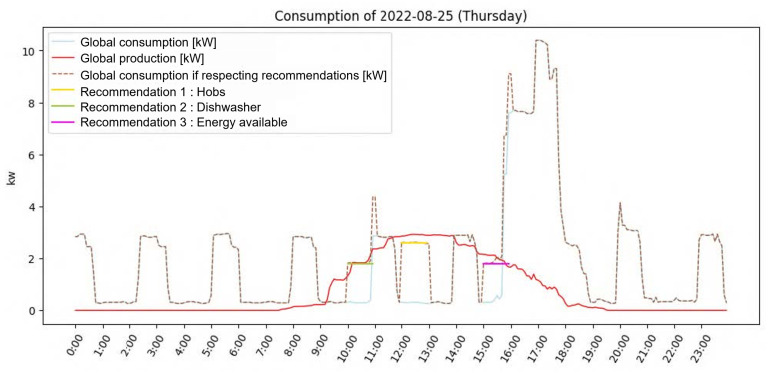
Graph illustrating the system fitting activities depending on the produced energy available.

**Figure 7 sensors-23-07974-f007:**
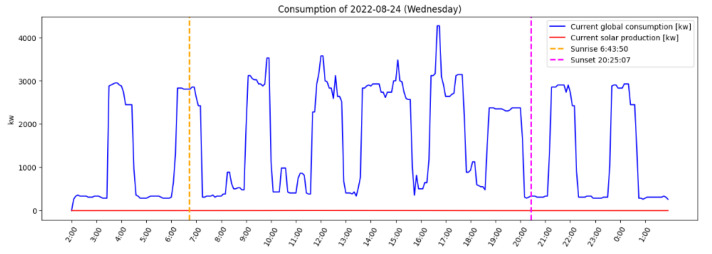
Example of graph presented in the evaluation document for respondents to validate the relevance of values contained in the textual recommendations. It shows the energy produced and consumed, along with sunrise and sunset times, and the ignition of devices in the house. This graph shows data collected on 16 August 2022.

**Table 1 sensors-23-07974-t001:** Summary of ML experiments conducted to predict the energy consumed in the house. *Prod*. = *production*; *PA* = *passive aggressive*; *SGD* = *stochastic gradient descent*; *MLP* = *multilayer perceptron*; *ETs* = *Extra-Trees*; *DT* = *decision tree*; *LR* = *linear regressor*.

ID	Features	Models	Shuffle	Ground Truth
EC1	Sensor logs	PA, SGD, MLP, ETs, DT, LR	Yes	Energy consumed instantly (5 min)
EC2	Weather + date	PA, SGD, MLP, ETs, DT, LR	Yes	Energy consumed instantly (5 min)
EC3	Weather + solar prod. + date	PA, SGD, MLP, ETs, DT, LR	No	Energy consumed instantly (5 min)
EC4	Weather + solar prod. + date	PA, SGD, MLP, ETs, DT, LR	No	Cumulative energy consumed
EC5	Weather + solar prod. + date	PA, SGD, MLP, ETs, DT, LR	No	Energy consumed instantly (1 h)

**Table 2 sensors-23-07974-t002:** Results obtained to predict the energy consumed in the house. The features reported here are those selected by the model that yielded the best score. *MLP* = *multilayer perceptron*; *ETs* = *Extra-Trees*; *LR* = *linear regressor*.

ID	Best Model	Selected Features	R${}^{2}$ Score	RMSE
EC1	MLP	Kitchen Spot, Lounge Chandelier 1, Kitchen Work Surface, month, time	9.61%	1.63
EC2	ETs	temp, humidity, windspeed, winddir, time	89.20%	0.57
EC3	ETs	temp, humidity, windspeed, winddir, current solar production, time	3.36%	1.75
EC4	LR	daily cumulated solar production, time	0.22%	1.63
EC5	ETs	temp, humidity, windspeed, winddir, current solar production, time	11.56%	1.13

**Table 3 sensors-23-07974-t003:** Summary of ML experiments conducted to predict the energy produced with solar panels. The features reported here are those selected by the model that yielded the best score. *PA* = *passive aggressive*; *SGD* = *stochastic gradient descent*; *MLP* = *multilayer perceptron*; *ETs* = *Extra-Trees*; *DT* = *decision tree*; *LR* = *linear regressor*.

ID	Features	Models	Shuffle	Ground Truth
EP1	Weather + date	PA, SGD, MLP, ETs, DT, LR	Yes	Energy consumed instantly (5 min)
EP2	Weather + date	PA, SGD, MLP, ETs, DT, LR	No	Energy consumed instantly (5 min)
EP3	Weather + date	PA, SGD, MLP, ETs, DT, LR	No	Energy consumed instantly (1 h)

**Table 4 sensors-23-07974-t004:** Results of the ML experiments to predict the energy produced with solar panels. *MLP* = *multilayer perceptron*; *ETs* = *Extra-Trees*.

ID	Best Model	Selected Features	R${}^{2}$ Score	RMSE
EP1	ETs	solar radiation, solar energy, uvindex, time	95.13%	0.20
EP2	MLP	temp, windspeed, solar radiation, time	84.81%	0.36
EP3	ETs	solar radiation, solar energy, uvindex, time	82.27%	0.37

**Table 5 sensors-23-07974-t005:** Nomenclature explaining variables in Equation (Equation 2).

Variable	Definition	Type/Unit/Range
Activity	Name of the activity recommended to the resident.	String (text).
ContextTime	Range of hours in which the user should carry out the above activity.	Array with 2 integers: start and end times.
Saving	The amount of energy saved if applying the recommendation.	Float, in kWh.
Likelihood	Probability that the user respects the recommendation.	Integer, between 0 and 100.
Ranking	The rank attributed to the recommendation.	Integer, from 1 (best) to 6 (worse).

**Table 6 sensors-23-07974-t006:** Results of the evaluation of the recommender system by the 11 respondents. *NA* = *Not Applicable*; *NO* = *No Occurrence*.

Textual Recommendation	Relevance	Variant 1		Variant 2	
		**Occurrences**	**Values**	**Occurrences**	**Values**
The production is excellent, so it would be an ideal day for cooking at midday. We have an output of at least 1900 W from 12 to 1 p.m., which is enough to run the hob(s) consuming about 1500 W for about 0.5 h.	100%	13 (23%)	75%	8 (30%)	75%
It’s a good day to start your dishwasher. Indeed, we have a production of at least 1500 W from 9 a.m. to 10 a.m., which is sufficient for this machine consuming about 1000 W for about 1 h.	100%	5 (9%)	76%	7 (26%)	79%
We produce enough to power appliances of around 1700 W from 10 a.m. to 11 a.m.	45%	15 (27%)	78%	3 (11%)	88%
Today’s weather is rather bad, so we’re not producing enough! If possible, plan your heavy consumption for another day.	82%	0 (0%)	NO	0 (0%)	NO
We predict that periodic consumption will consume about 3 kW for about an hour at these times: 2 a.m., 5 a.m., 8 a.m., 10 a.m., 1 p.m., 4 p.m., 7 p.m., 10 p.m., 0 a.m.	36%	14 (25%)	38%	NA	NA
It’s going to be hot today, so you could save energy by turning down the heating.	91%	3 (5%)	NA	3 (11%)	NA
It’s going to be cold today, and we’re predicting a possible increase in heating consumption. Remember not to open the windows for too long to ventilate: 2 min are enough with the room door open, 10 min with the door closed and 1 h with the window also transomed.	91%	0 (0%)	NA	0 (0%)	NA
On average, a 10-min shower consumes 50 L of water, which requires 2.3 kW to heat.	64%	2 (4%)	NA	2 (7%)	NA
Glass ceramic hobs have a power rating of 2000 W, while induction hobs have a power rating of 2800 W. On the other hand, induction hobs heat up faster, which means they use less energy on average.	27%	1 (2%)	NA	1 (4%)	NA
Yesterday, a high consumption occurred in the afternoon when your solar panels were not producing. If possible, place your high consumption on days when the panels are producing.	82%	0 (0%)	NO	0 (0%)	NO
Yesterday, high consumption occurred in the afternoon when your panels were producing. This is a very good practice, keep it up!	91%	3 (5%)	64%	3 (11%)	64%
* **Average/Sum** *	**74%**	**56 (100%)**	**66%**	**27 (100%)**	**77%**

## Data Availability

The data collected for this work is not available.

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
