# Peer review of "A Recommender System for Increasing Energy Efficiency of Solar-Powered Smart Homes"

_sensors, 2023, doi:10.3390/s23187974_

Round 1

Reviewer 1 Report

·         The Abstracts must contain at least 150 words up to 250 words, consist of

2-3 sentences as brief intro about the paper, 2-3 sentences to describe how

the problem is solved, and 2-4 sentences showing the results of experiments/simulation, ended with 1-2 sentences as short main conclusions of the work.

·         The Introduction section must explain the background of the problem and the urgency of the study, which can be proved by providing some previous researches and works, and also how to solve the problem in brief.

·         Novelty not clear. Add to last paragraph in introduction.

·         In general, you must provide the units of the variables and symbols used in the equation. A table of Nomenclature (with definition and units) can be used or you can explain the symbols and units in the text.

·         Figure captions should be standalone, i.e., descriptive enough to be understood without having to refer to the main text.

·         Figures not clear.

·         The conclusion should be more precise, showing only the main results.

·          Better description and explanation of results.

·         Added future work in conclusion.

·         Comparing with previous work.

·         Update reference 2018 to 2023.  

Author Response

We would like to thank you for your valuable feedbacks on our manuscript. We addressed most of them and we hope that this improved version of the manuscript now meets your requirements. The modifications made to the manuscript are detailed below:

  1. The abstract has been slightly improved:
    1. Formulation of sentences was improved and more concise
    2. 1-2 sentences with short main conclusions of the work were added at the end
    3. It fits in the 250 words required and contains all sections mentioned
  2. We agree that the novelty of the proposed system was not clearly mentioned. This has been addressed at the end of the introduction: L59-65.
  3. A nomenclature was added to explain the equation (2). Table 5 was added and introduced in L310.
  4. For the figures, we believe they are standalone and can be understood without refereeing to the text. Could you specify which one(s) are not standalone and/or clear in your opinion? Thank you for the precisions.
  5. In the conclusion, we agree that the main findings have not been highlighted and that the most important points to be addressed in future work have not been mentioned. This has been corrected as the conclusion has been revised: L487-501. For your concern, we are already working on several of these proposals to develop a second version of the system that addresses the limitations found in the first version.
  6. We believe that the results are already enough detailed (around 3 pages out of 15, without appendices and references). If you think that a specific result or section should be better detailed, could you please specify which one? Thank you.
  7. To be able to compare our results with previous work regarding the prediction of energy consumption and production, we reported the RMSE achieved in each experiment (see Tables 2 and 4). The results of ML predictions are now compared and discussed in section 5.2 (L404-412 for energy consumption and L416-424).
  8. For the years of publication, we believe it was most important to reference recent publications in section 2.1 about the prediction of energy consumed and produced with AI. This was done as most of publications cited in this section [6 to 14] were published after 2018. In section 2.2, we thought it is important to report all publications proposing similar recommender systems than the one proposed in this manuscript, regardless their publication year. We hope you will agree with our choice. Still, 3 references out of 4 cited in section 2.2 were published in 2020 or later.

Reviewer 2 Report

The paper titled "A Recommender System for Increasing Energy Efficiency of Solar-Powered Smart Homes" presents the development and evaluation of a recommender system designed to optimize the energy produced by solar panels in smart homes. The system utilizes artificial intelligence and machine learning algorithms to predict energy production and consumption based on sensor logs, solar inverter data, and weather information. Recommendations are generated and ranked based on their relevance to the user. The study evaluates the system by testing the recommendations' relevance, ranking, and diversity with 11 participants.

The article presents a very thorough analysis of the impact of the models in reducing the energy waste and create the best recomendation for the end users.

The future work mentions the usage of of eXplainable Artificial Intelligence (XAI) methodologies. Employing XAI techniques offers an opportunity to delve into the intricate behaviors exhibited by solar panel sensors, thereby fostering a deeper and more transparent understanding of their underlying mechanisms. Some suggestions for the authors can employ a global explanation of the best model to better grasp what makes the energy expenses increase or decrease based on the sensor data (e.g, Using SHAP or IMPORTANCE in Rminer), or even explain specific outliers (e.g., using local explainers such as LIME or IMPORTANCE in Rminer)

By employing XAI, the study can also generate of diverse scenarios devoid of necessitating direct end user inputs (e.g., What-IF analysis). This approach enables the exploration of predictive model outcomes within a versatile spectrum of circumstances, culminating in a richer analysis of sensor behavior.

This could be included in this work if there are no limitations in the journal or article since there are several packages/softwares that assist in obtaining these results (depending on the model).

I hope my comments are useful to improve the quality of the paper.

Author Response

Thank you for your feedback on the manuscript. Actually, this is the first version of the recommender system and we rather planned of including xAI in a second version of the system on which we are already working on. We are planning to test different explanation modalities (no explanation, textual explanation, textual explanation + graph). To address your remarks, 3 feature selection techniques were tested with all combination of features sets were tested in this first system version. The manuscript already highlights which combination of features led to the best score for each experiment. To be more explicit, we included in the caption the caption of tables 2 and 4 that the features reported are those selected by the model that yielded the best score. If you really think we should go further in the explanation at this step, we could report the score achieved by each feature in selection process, and order them in descending order for instance.

Round 2

Reviewer 1 Report

publish